# Long term functional plasticity of sensory inputs mediated by olfactory learning

**Nixon M Abraham[1,2]*[†], Roberto Vincis[1,2][†], Samuel Lagier[1,2], Ivan Rodriguez[2,3], Alan Carleton[1,2]***

[1]Department of Basic Neurosciences, School of Medicine, University of Geneva, Geneva, Switzerland; [2]Geneva Neuroscience Center, University of Geneva, Geneva, Switzerland; [3]Department of Genetics and Evolution, University of Geneva, Geneva, Switzerland

**Abstract** Sensory inputs are remarkably organized along all sensory pathways. While sensory representations are known to undergo plasticity at the higher levels of sensory pathways following peripheral lesions or sensory experience, less is known about the functional plasticity of peripheral inputs induced by learning. We addressed this question in the adult mouse olfactory system by combining odor discrimination studies with functional imaging of sensory input activity in awake mice. Here we show that associative learning, but not passive odor exposure, potentiates the strength of sensory inputs up to several weeks after the end of training. We conclude that experience-dependent plasticity can occur in the periphery of adult mouse olfactory system, which should improve odor detection and contribute towards accurate and fast odor discriminations.

**\*For correspondence:** nixon.
abraham@unige.ch (NMA); alan.
carleton@unige.ch (AC)

[†]These authors contributed
equally to this work

**Competing interests:** The
authors declare that no
competing interests exist.

**Reviewing editor**: Howard
Eichenbaum, Boston University,
United States

## Introduction

Mammalian brains remain plastic throughout their life span, enabling animals to adapt their behavior to novel conditions. Both structural and functional plasticity have been reported in different sensory systems during development and in adulthood following lesions, passive experience or learning (*Recanzone et al., 1993*; *Bao et al., 2001*; *Polley et al., 2004*; *Accolla and Carleton, 2008*; *Keck et al., 2008*; *Hofer et al., 2009*; *Carleton et al., 2010*; *Rosselet et al., 2011*; *Lai et al., 2012*). While attention has been mostly focused on plasticity mechanisms in higher brain areas (i.e., mainly in the cortex) (*Buonomano and Merzenich, 1998*; *Bao et al., 2001*; *Feldman and Brecht, 2005*; *Accolla and Carleton, 2008*; *Carleton et al., 2010*; *Rosselet et al., 2011*; *Lai et al., 2012*), less is known about the functional changes at earlier stages of sensory processing, especially in response to sensory learning. Here we investigated the functional plasticity of sensory inputs in the mouse olfactory system.

Olfactory sensory neurons (OSNs) expressing one specialized odorant receptor gene out of a large repertoire (*Malnic et al., 1999*) converge in a receptor-specific manner onto anatomical structures in the main olfactory bulb (OB) called glomeruli (*Ressler et al., 1994*; *Mombaerts et al., 1996*; *Sakano, 2010*). Odorants activate complex spatio-temporal patterns of glomeruli, which can be monitored with various imaging techniques (*Rubin and Katz, 1999*; *Uchida et al., 2000*; *Wachowiak and Cohen, 2001*; *Spors and Grinvald, 2002*; *Bozza et al., 2004*; *Bathellier et al., 2007*; *Vincis et al., 2012*; *Patterson et al., 2013*). The sensory information received in the glomeruli by OB output neurons is then transferred to cortical areas.

Several plasticity mechanisms have been reported in olfactory cortical regions (*Quinlan et al., 2004*; *Franks and Isaacson, 2005*; *Stripling and Galupo, 2008*) as well as OB circuitry (*Saghatelyan et al., 2005*; *Marks et al., 2006*; *Gao and Strowbridge, 2009*; *Livneh and Mizrahi, 2012*). At the input level, both sensory deprivation (*Cummings et al., 1997*) and developmental reorganization (*Zou et al., 2004*; *Kerr and Belluscio, 2006*) have been reported to induce structural plasticity in the

**eLife digest** The mammalian brain is not static, but instead retains a significant degree of plasticity throughout an animal's life. It is this plasticity that enables adults to learn new things, adjust to new environments and, to some degree, regain functions they have lost as a result of brain damage.

However, information about the environment must first be detected and encoded by the senses. Odors, for example, activate specific receptors in the nose, and these in turn project to structures called glomeruli in a region of the brain known as the olfactory bulb. Each odor activates a unique combination of glomeruli, and the information contained within this 'odor fingerprint' is relayed via olfactory bulb neurons to the olfactory cortex.

Now, Abraham et al. have revealed that the earliest stages of odor processing also show plasticity in adult animals. Two groups of mice were exposed to the same two odors: however, the first group was trained to discriminate between the odors to obtain a reward, whereas the second group was passively exposed to them. When both groups of mice were subsequently re-exposed to the odors, the trained group activated more glomeruli, more strongly, than a control group that had never encountered the odors before. By contrast, the responses of mice in the passively exposed group did not differ from those of a control group.

Given that the response of glomeruli correlates with the ability of mice to discriminate between odors, these results suggest that trained animals would now be able to discriminate between the odors more easily than other mice. In other words, sensory plasticity ensures that stimuli that have been associatively learned with or without reward will be easier to detect should they be encountered again in the future.

glomerular layer. Despite these facts, little is known about learning-mediated functional plasticity of sensory inputs in the adult OB of awake mice.

Here, we investigate plasticity at the periphery induced by learning or passive exposure by combining olfactory behavior and functional imaging in awake mice. Olfactory training caused an enhanced sensitivity and potentiation of sensory inputs, which helped the animals to achieve fast and accurate odor discrimination. Most strikingly, this functional plasticity was induced specifically by the learning process but not by a passive exposure to the same odorants, and lasted up to several weeks.

## Results

### Defining odor discrimination thresholds using a wide range of odorant dilutions

To investigate the potential functional plasticity at the level of sensory neurons induced by olfactory learning, mice were trained to discriminate two odorants (rewarded vs unrewarded) on a go/no-go operant conditioning paradigm (*Abraham et al., 2004*, *2010*). As perception can vary with the odorant dilution, we used a wide spectrum of dilutions covering several orders of magnitude ranging from $10^0$ to $10^{-10}$ (percentile dilution in mineral oil) for two different odor pairs (cineol [Cin] vs eugenol [Eu] and isoamyl acetate [IAA] vs ethyl butyrate [EB]) (*Figure 1A*). After training, odorant-evoked input patterns were measured in the olfactory bulb by intrinsic optical signal (IOS) imaging (*Figure 1B,C*). To control for the effect of odorant exposure during olfactory discriminative learning, two other groups of mice were also imaged (naïve and passively exposed groups, *Figure 1D*). Naïve mice never encountered the odorants used for behavior testing before the imaging session whereas the passively exposed group of mice received the same amount of trials and stimulus exposure than the trained group.

After a habituation phase ('Materials and methods'), mice were trained to discriminate Cin vs Eu ($1200$ trials at $10^0$, *Figure 2A*). Performance levels reached more than 80% of correct responses after 400 trials and remained high during the following sessions. This task allowed the mice to acquire the procedural aspects of the training. It also allowed us to test the discrimination abilities for this odor pair at low dilutions (i.e., high concentrations). We then tested the discrimination abilities for different dilutions of the same odor pair. The performance levels remained close to chance levels for $10^{-10}$ and $10^{-5}$ dilutions, which can be interpreted as a lack of perception/discrimination

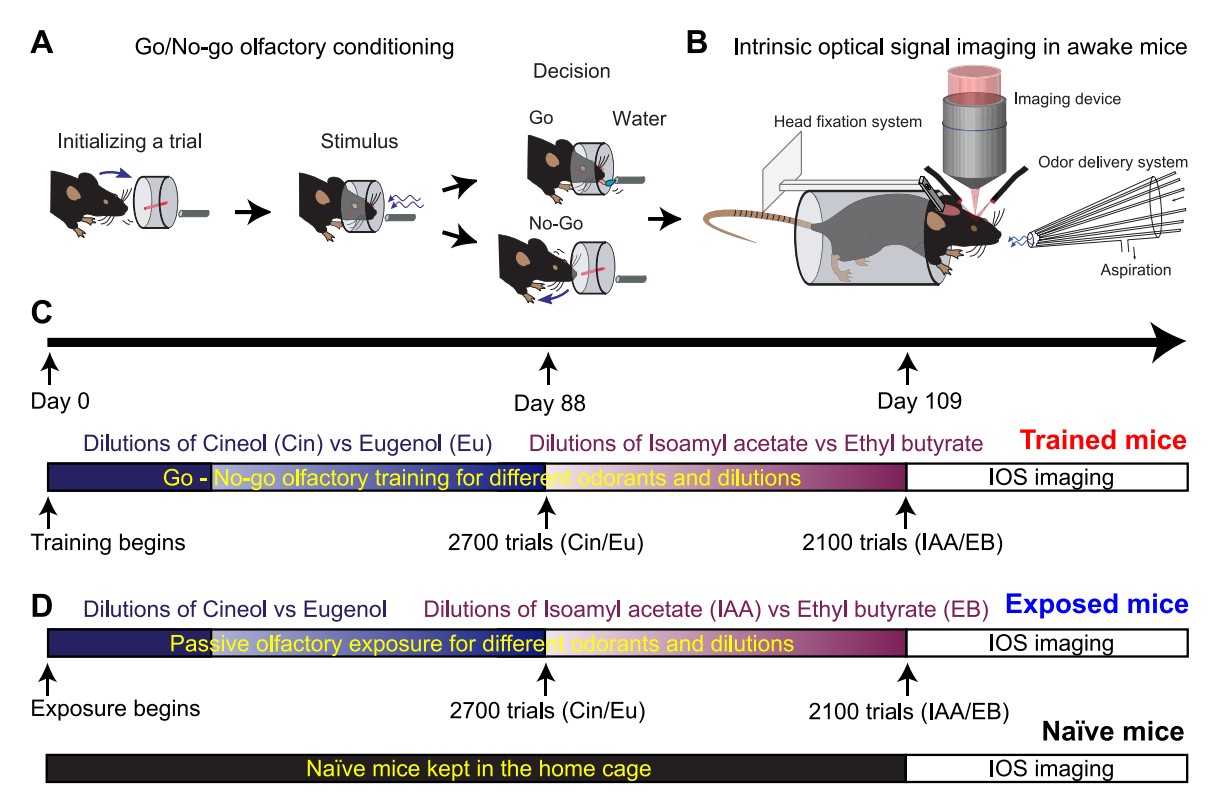

**Figure 1**. Experimental design used to assess plasticity of sensory inputs in the mouse OB. (**A** and **B**) Mice are first trained to discriminate pairs of odorants in an automated olfactometer. At the end of the training, odorants evoked input patterns are monitored on the dorsal OB of awake mice using intrinsic optical signal imaging. (**C**) Timetable of the go/no-go olfactory training for different odorants at different dilutions and then followed by imaging (Trained group). (**D**) Two other groups of mice have been imaged for comparison. Several mice have been passively exposed to the different odorants used for the training and for the same amount of time (Exposed group). A third group of mice that never experienced the odorants served as control (Naïve group).

(*Figure 2A*, no difference between the two dilutions, Fisher's Least Significant Difference [LSD] test, p>0.05). On average, mice started to discriminate from $10^{-3}$ onwards (note that one mouse was able to discriminate at $10^{-5}$ with an accuracy of ~70%, *Figure 2B*), reaching ~70% of correct responses for $10^{-3}$ dilution and >90% for $10^{-2}$, $10^{-1}$ and $10^{0}$ dilutions (*Figure 2B*). Following the Cin vs Eu training, mice were trained for different dilutions of IAA vs EB in the range of $10^{-10}$-$10^{0}$. Performance levels remained close to chance levels for $10^{-10}$ and $10^{-8}$ dilutions (no difference between the two dilutions, LSD, p>0.1, paired comparison), but mice showed a tendency to learn at $10^{-6}$ dilution and hence were trained at this dilution for an additional 300 trials during which their performance reached 80% of correct responses (comparison between first and second task of 300 trials $10^{-6}$ IAA/EB, LSD test p<0.005). For lower dilutions ($10^{-4}$, $10^{-2}$ and $10^{0}$), mice performed systematically above 90% (*Figure 2C*).

As we observed high and stable performance levels at different dilutions, we calculated the reaction times, a more sensitive parameter to monitor discrimination behavior (*Abraham et al., 2004, 2010, 2012*). Across all concentrations, reaction times decreased with the odorant dilution, reflecting a direct linear correlation with performance levels ($R^2$ = 0.74, ANOVA *F* = 12.3 p=0.04 and $R^2$ = 0.82, ANOVA *F* = 23.2 p=0.0085 for Cin/Eu and IAA/EB tasks, respectively). For dilutions with high performance levels, reaction times were relatively stable. Mice discriminated the dilutions $10^{-2}$, $10^{-1}$ and $10^{0}$ of Cin vs Eu with similar reaction times (*Figure 2D*, one-way repeated measures ANOVA, *F* = 2.84, p=0.13). Different dilutions of IAA vs EB ($10^{-4}$, $10^{-2}$ and $10^{0}$) were also discriminated with similar speeds, though significantly different (*Figure 2E*, one-way repeated measures ANOVA, *F* = 3.96, p=0.035; Post-hoc LSD test p=0.043, 0.016 and 0.63 for $10^{0}$ vs $10^{-2}$, $10^{0}$ vs $10^{-4}$ and $10^{-2}$ vs $10^{-4}$, respectively).

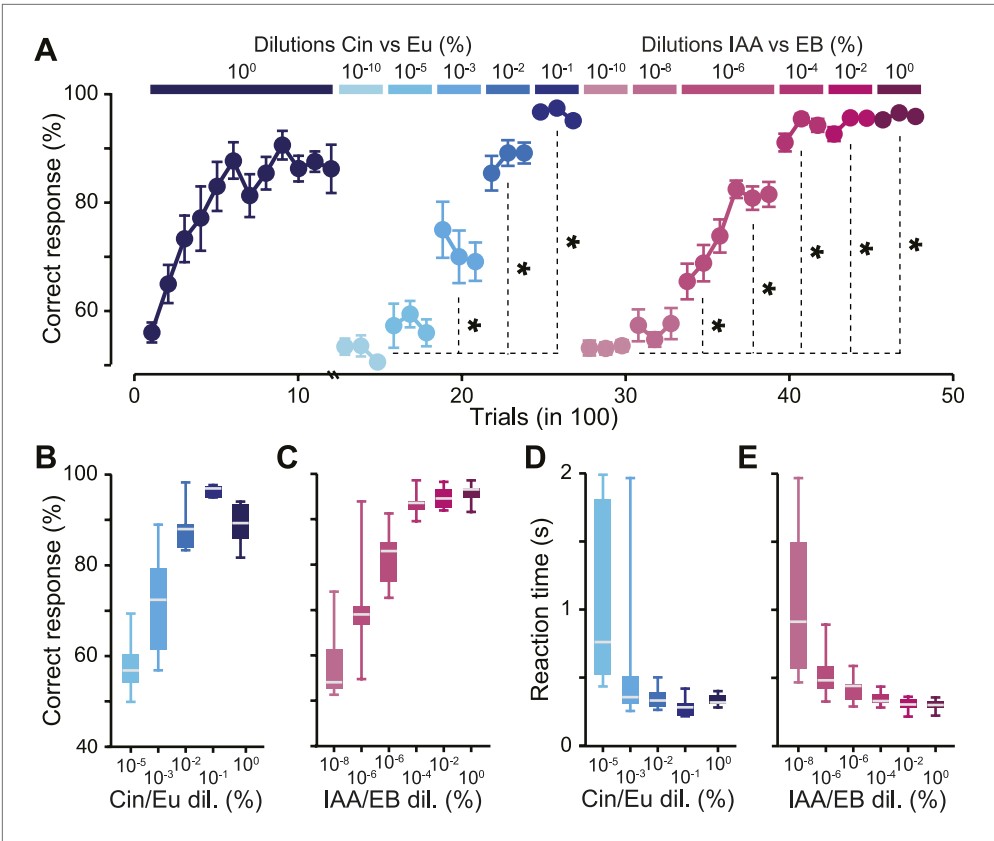

**Figure 2**. Defining odor discrimination threshold using wide range of odorant dilutions. (**A**) Discrimination accuracy shown as the average percentage of correct choices for different odorants over wide range of dilutions ($n = 7$ and 11 mice for Cin/Eu and IAA/EB tasks, respectively). The population of mice showed a tendency for learning to discriminate Cin/Eu from $10^{-3}$ onwards and IAA/EB from $10^{-6}$ onwards [*: Fisher's Least Significant Difference (LSD) test at least p<0.005]. Data are presented as mean ± SEM. (**B** and **C**) Discrimination accuracy measured on the last 300 trials for different dilutions of Cin/Eu and IAA/EB. (**D** and **E**) Reaction time (RT) measured on the same trials as in (**B** and **C**). Data are presented as box plots showing the median in gray. Whiskers represent the maximum and minimum values of the dataset.

In summary, mice were able to discriminate a broad range of dilutions with similar accuracy and speed above certain odorant dilutions, defining discrimination thresholds. For the population of trained animals, we therefore estimate that the discrimination thresholds for Cin/Eu and IAA/EB are between $10^{-5}$ and $10^{-3}$ and between $10^{-8}$ and $10^{-6}$, respectively.

## Olfactory learning induces functional plasticity at the level of sensory neuron inputs

We then investigated the effect of olfactory learning on sensory input representation. In order to assess functional plasticity of the inputs, we monitored odorant-evoked glomerular patterns on the dorsal OB of trained mice, with IOS imaging in awake head-restrained animals (*Vincis et al., 2012*). For comparison, we used naïve mice that had never experienced the odorants and a group of mice that had been passively exposed to the same odorants and dilutions used for training (*Figure 1D*).

For odorant dilutions lower than the discrimination threshold (≤$10^{-5}$ and $10^{-8}$ for Cin/Eu and EB/IAA, respectively), we could not detect any activated glomeruli on the dorsal OB of any group of mice (*Figure 3A–E*). For odorant dilutions above the discrimination threshold ($10^{-3}$ to $10^{-2}$ and $10^{-6}$ to $10^{-4}$ for Cin/Eu and EB/IAA, respectively; referred as high dilution hereafter), we observed more activated glomeruli in trained animals than in naïve or exposed mice, an effect consistent across all odorants used for training (*Figure 3A–E*). On average we observed a threefold increase in the average number

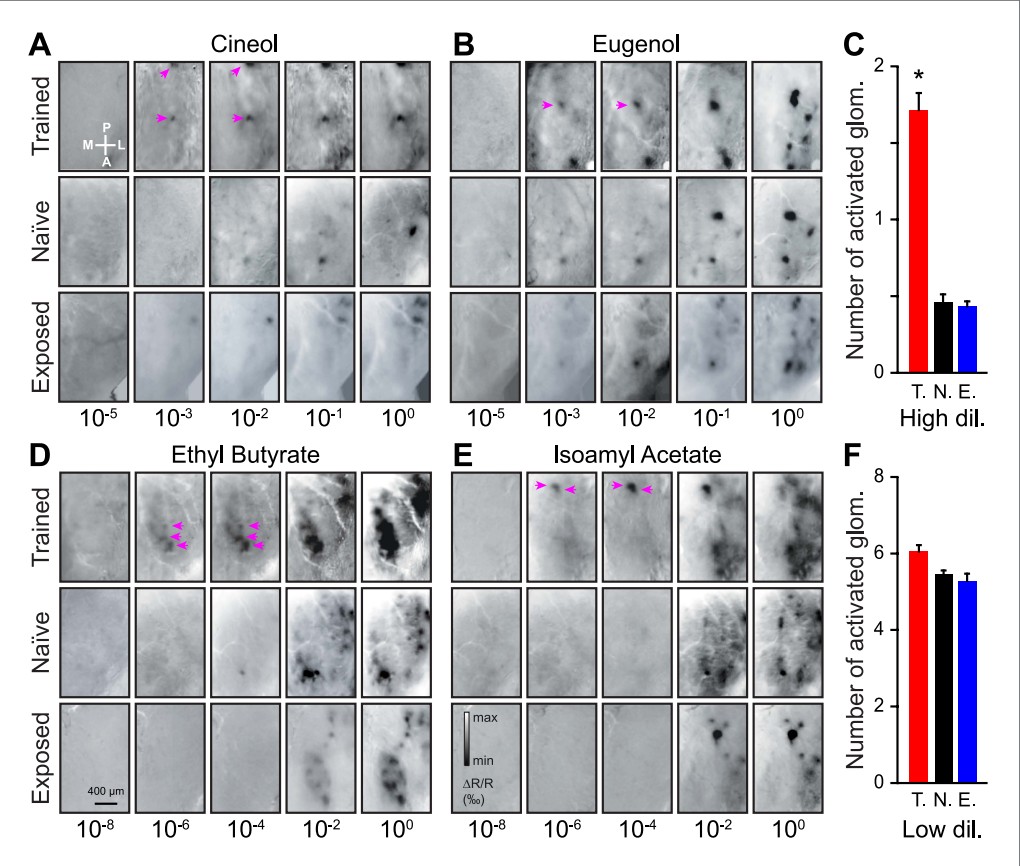

**Figure 3**. Functional plasticity of sensory inputs to the olfactory bulb induced by olfactory learning. (**A** and **B**) Intrinsic optical signal (IOS) imaging of activated glomeruli patterns evoked by different dilutions of Cin/Eu in different groups of mice (trained, naïve and exposed). For each odorant and group, all images for the different dilutions are from the same mouse. For the scale in (**A**) and (**B**), respectively, min $\Delta R/R$ (‰) = −2 and −2.5, max $\Delta R/R$ (‰) = 1.5 and 2. The magenta arrows highlight that more strongly activated glomeruli are visible in the trained mice at lower dilutions. (**C**) The average number of glomeruli activated by all odorants at higher dilutions ($10^{-3}$ and $10^{-2}$ for Cin/Eu, $10^{-6}$ and $10^{-4}$ for IAA/EB) is significantly higher in the trained group (T., $n = 5$ mice; LSD test between T. and N. or E.: p<0.001) than in the naïve (N., $n = 5$ mice) or exposed (E., $n = 5$ mice; LSD test between E. and N. p=0.5) groups. (**D** and **E**) IOS imaging of the activated glomeruli patterns evoked by different dilutions of IAA/EB. (**F**) The average number of glomeruli activated by all odorants at lower dilutions ($10^{-1}$ and $10^{0}$ for Cin/Eu, $10^{-2}$ and $10^{0}$ for IAA/EB) is similar for all groups (for all comparisons, LSD test p>0.1). For the scale in (**D**) and (**E**), respectively, min $\Delta R/R$ (‰) = −3.5 and −3, max $\Delta R/R$ (‰) = 3 and 2.5. Values are represented as mean ± SEM.

of activated glomeruli among the trained mice (*Figure 3C*). This enhancement was due to the associative learning and not due to simple exposure to the odorants since the animals passively exposed to the same stimuli and for the same amount of time did not show a significant change in the number of glomeruli in comparison to naïve mice (*Figure 3C*).

The observed increase in the number of glomeruli could be due to an enhanced sensitivity of the glomeruli normally activated at lower dilutions or due to the recruitment of new glomeruli. At lower dilutions ($10^{-1}$ to $10^{0}$ and $10^{-2}$ to $10^{0}$ for Cin/Eu and EB/IAA, respectively; referred as low dilution hereafter) the number of activated glomeruli remained similar across the three groups (*Figure 3F*), suggesting that associative learning indeed improves the sensitivity of sensory inputs.

We then asked if there were any changes in the odorant-evoked activity at lower dilutions. In order to address this point we quantified the amplitude of glomerular responses (change in reflectance) for all learned dilutions/odorants. The amplitude of the evoked activity did not differ among the passively exposed and naïve group of mice (*Figure 4A–D*, LSD test p=0.7, 0.23, 0.81 and 0.2 for IAA, EB, Cin

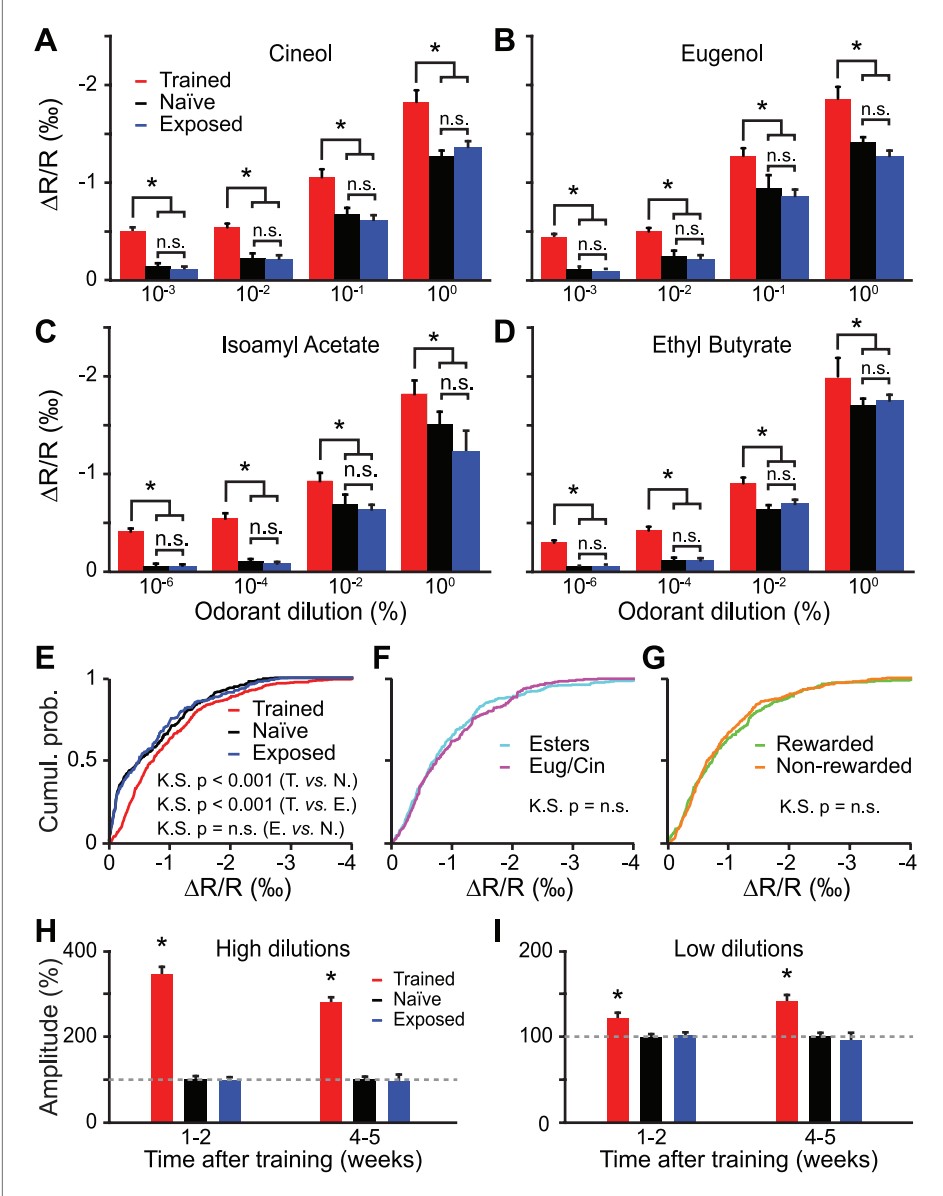

**Figure 4**. Functional plasticity induced by olfactory learning is long lasting and independent of the reward value of odorants. (**A–D**) Quantification of the average change in reflectance (ΔR/R) in the glomeruli activated by different odorants and dilutions (*n* = 5 mice for all groups, * indicates LSD test p<0.001). (**E**) Cumulative distributions of amplitudes of the evoked activity in all glomeruli analyzed in all trained (*n* = 512 regions of interest), naïve (*n* = 431) and exposed (*n* = 427) mice. A significant [Kolmogorov–Smirnov (K.S.) test] increase in amplitude is observed for the population of glomeruli recorded in the trained group. (**F-G**) In the trained group, no difference in response amplitude was observed between glomeruli activated by different odorants belonging to different chemical classes (**F**) or between glomeruli activated by rewarded and non-rewarded odorants (**G**). (**H** and **I**) The training is potentiating the input strength for several weeks. All values have been normalized to the average amplitude calculated in the naïve group. *: LSD test between trained and other groups at least p<0.002.

and Eu). In contrast, compared to these control groups, trained mice consistently showed enhanced IOS amplitudes (*Figure 4A–D*). In summary, a form of functional plasticity is induced in OB sensory inputs by an olfactory learning, but not by a passive exposure to the same odorants. This potentiation was independent of the chemical class and the dilution of the stimuli (*Figure 4E,F*).

## Functional plasticity induced by olfactory learning is long-lasting and independent of the reward value

In the go/no-go operant conditioning paradigm used for olfactory training, one stimulus is associated with a reward, whereas the second stimulus is neither punished nor rewarded. It is therefore possible that the sensory representation of the rewarded stimulus is more strongly potentiated compared to the non-rewarded one. However the amplitude of the glomerular responses evoked across all dilutions was not significantly different between the rewarded and non-rewarded stimuli (*Figure 4G*).

During the behavioral experiments, we started the training with different dilutions of Cin/Eu followed by dilutions of IAA/EB (*Figure 1C*). This resulted in different post-training delays before recording evoked IOS for different odor pairs (4–5 weeks for Cin/Eu and 1-2 weeks for IAA/EB) and allowed us to study whether the functional plasticity is long lasting or not. We compared the amplitude of the evoked glomerular responses across the three groups and found that the learning-induced potentiation is visible for all dilutions and lasts up to 5 weeks (*Figure 4H,I*).

In conclusion, olfactory learning induces a form of functional plasticity at the sensory input level, which is long lasting and independent of the stimulus reward value.

## Functional plasticity enhances the discriminability of odorants in the perception range close to their discrimination threshold

Olfactory learning induces the potentiation of sensory inputs to the OB, increasing both the number of activated glomeruli and the strength of activation when compared to control animals. In order to investigate the behavioral relevance of this form of plasticity, we plotted the relationship between discrimination performance levels and measured glomerular response amplitudes for different odorant dilutions. These data were well fitted with a Boltzmann function for both discrimination tasks (*Figure 5A*). In the rising phase of the curves, corresponding to the dilutions around discrimination threshold, a small change in input strength caused strong improvement in discrimination accuracy. This was further verified by comparing the animals' accuracy at dilutions close to their discrimination threshold with the strength of OB inputs. At these specific dilutions, an increase in the performance level (*Figure 5B*) correlated with an increase in input strength; either by an increase in the number of activated glomeruli or by an increase in the amplitude of glomerular responses (*Figure 5C,D*).

Since we observed that the learning process potentiated the inputs to the olfactory bulb, we propose that the trained animals would be able to detect and eventually discriminate more easily and rapidly odorants at lower concentrations than control animals (i.e., naïve and exposed groups). The discrimination threshold for odorants could potentially be shifted by few orders of concentration magnitude.

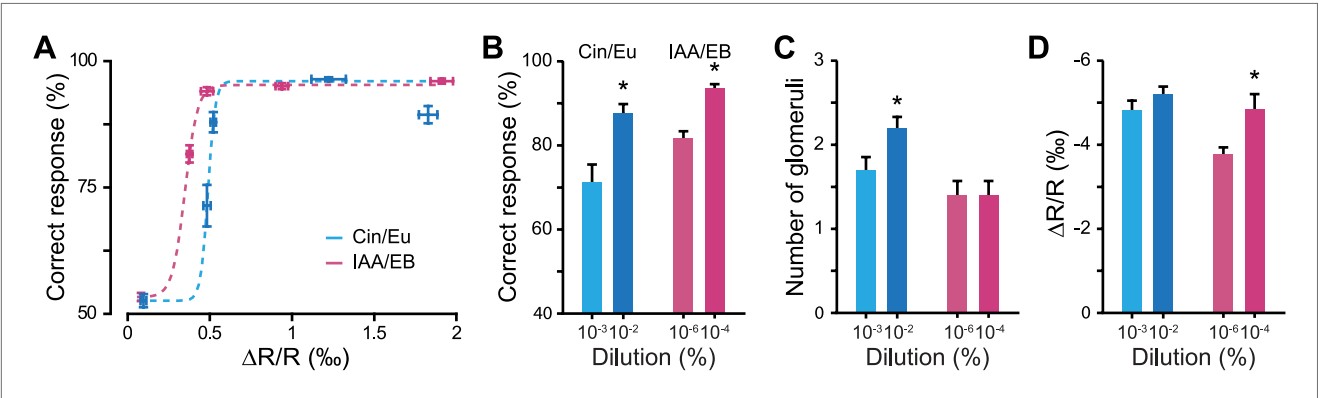

**Figure 5**. Discrimination accuracy around discrimination threshold is dependent on input strength. (**A**) Plots showing the relationship between the average input strength monitored by IOS imaging and the discrimination accuracy at different concentrations for different odor pairs. The dotted lines represent Boltzmann function fits in the distributions of points (Boltzmann fit, $R^2 > 0.95$, $F = 511.34$, ANOVA $p<0.032$). (**B**) Discrimination accuracy for odorant concentrations close to discrimination threshold. *: Paired *t* test: $10^{-3}$ vs $10^{-2}$ Cin/Eu, $p<0.01$ and $10^{-6}$ vs $10^{-4}$ IAA/EB, $p<0.01$. (**C**) Average number of activated glomeruli for odorant concentrations close to discrimination threshold. *: Paired *t* test: $10^{-3}$ vs $10^{-2}$ Cin/Eu, $p<0.05$ and $10^{-6}$ vs $10^{-4}$ IAA/EB, $p>0.1$. (**D**) Quantification of the average change in reflectance (ΔR/R) in the glomeruli activated by odorants at concentrations close to discrimination threshold. *: Paired *t* test: $10^{-3}$ vs $10^{-2}$ Cin/Eu, $p>0.1$ and $10^{-6}$ vs $10^{-4}$ IAA/EB, $p<0.01$.

## Discussion

Our study reveals the existence of a functional plasticity at an early stage of the mouse olfactory pathway. An associative learning, but not a passive exposure to the same odorants, increases the odorant detection sensitivity (*Figure 3*) and induces a potentiation of the input strength at all odorant dilutions (*Figure 4*). This potentiation was independent of the reward value and of the nature of the stimuli (*Figure 4*). Most strikingly, this form of plasticity lasted up to 5 weeks after the training (*Figure 4*), which could implicate such form of plasticity in long lasting improvement of perception abilities (*Figure 5*).

### Olfactory learning induces long-lasting potentiation of sensory inputs

The reported form of functional plasticity, induced by sensory experience, in the glomerular layer of the adult mouse olfactory system differs from previously reported plastic changes observed during the postnatal maturation of the olfactory system (*Zou et al., 2004*; *Kerr and Belluscio, 2006*). During development, OSNs expressing different receptors can project to the same glomerulus. During the first 2 months of postnatal life (*Zou et al., 2004*), a refinement of the projections occurs, leading to the known concept of a glomerulus receiving only afferents from sensory neurons expressing the same receptor. This form of structural plasticity is accelerated by sensory experience (*Kerr and Belluscio, 2006*), but it is not known if this causes any change in sensory input strength. In our study, the observed plasticity is induced during adulthood outside this developmental window, when one glomerulus is homogeneously innervated by one type of sensory neurons, and is associated to the increase of input strength.

All forms of sensory experience are not equivalent in triggering plasticity of the sensory inputs. Indeed, we report that associative learning causes a form of plasticity whereas a passive repetitive exposure did not, as similarly observed in cortical areas of other sensory systems (*Bao et al., 2001*; *Accolla and Carleton, 2008*; *Rosselet et al., 2011*; *Lai et al., 2012*). The passively exposed group of mice did not show any change in the input strength compared to the naïve ones. Differences between associative learning and passive exposure paradigms may be due to differences in the final concentration and presentation duration of the applied odorants. Though we cannot rule out this possibility, the lack of effect during passive exposure is consistent with a recent study (*Kato et al., 2012*). On the other hand, it is contrasting with the previously reported enhancement of input sensitivity following long passive exposure (*Wang et al., 1993*). In our study the total exposure time was less than an hour for one class of chemical stimuli (including all dilutions) whereas the exposure time was much longer in the latter study (16 hr daily during 2–6 weeks). In addition, the change in sensitivity reported previously was only observed for a barely detectable odorant in a selected mouse strain and not for other odorants. Therefore this non-generalized change in sensitivity might have resulted from extremely long exposure time for special odorants.

Our associative learning paradigm potentiated the input strength for several odorants independently of their chemical class and reward value. Previous reports showed that training specifically affected the odorant representation of the rewarded stimulus (*Faber et al., 1999*) or the stimulus associated with a foot shock (*Kass et al., 2013*). In contrast, we saw a similar potentiation for all odorants regardless of the reward value of the stimuli. What is the possible role of the reward value in causing the potentiation we observed? In the go/no-go discrimination-learning paradigm, one stimulus is coupled with a positive reward, whereas the second stimulus is neither punished nor rewarded. This task involves the decision-making process for both rewarded and non-rewarded odors, which involves the assessment of reward value. This may explain the learning-induced potentiation we observed for rewarded as well as non-rewarded odors. The enhancement of sensory input strength for both odorants could activate the inhibitory neurons of the olfactory bulb thereby helping the refinement and the discrimination between odorants (*Abraham et al., 2010*).

### Possible mechanisms underlying the long-lasting plasticity and implications for behavior

IOS imaging is primarily monitoring neurotransmitter release from OSNs as evidenced by pharmacology experiments (*Gurden et al., 2006*). Although we cannot completely rule out a possible postsynaptic contribution to the signal, studies comparing IOS with presynaptic-specific imaging readouts provide additional evidence supporting the presynaptic origin of IOS signals in the OB

(**Wachowiak and Cohen, 2003**; **Soucy et al., 2009**). In addition, a similar potentiation of the odor-evoked glomeruli activity induced by fear learning has been recently reported in anesthetized animals while monitoring a genetically encoded reporter of activity expressed specifically in OSNs (**Kass et al., 2013**). Taken together, this suggests that the plasticity is associated with change in OSN activity.

Which mechanism could explain a long-term change of OSNs activity? Firstly, OSN sensitivity could be enhanced, leading to an increase of the odor evoked OSN firing rate causing more release of glutamate in the glomerulus. Secondly, olfactory training could modify OSN turnover (**Schwob et al., 1992**) leading to higher number of axons innervating the same glomerulus (**Jones et al., 2008**). Thirdly, a local network mechanism could alter OSN release. Since activation of $GABA_B$ (**Aroniadou-Anderjaska et al., 2000**; **McGann et al., 2005**) and dopamine $D_2$ (**Hsia et al., 1999**; **Ennis et al., 2001**) receptors on OSN terminals can decrease glutamate release, inhibition of GABA and/or dopamine release would lead to the observed increase of input strength. Finally, a direct increase in glutamate release mediated by neuromodulatory fibers may also account for the observed plasticity. Further experiments are needed to identify the exact mechanisms underlying the long lasting change in input strength induced by this olfactory learning.

As we performed imaging experiments in awake mice, we took respiration behavior into account while analyzing the IOS readout. Previous studies have shown how an animal can increase its breathing frequency when sampling an odor (**Verhagen et al., 2007**; **Carey et al., 2009**; **Wesson et al., 2009**; **Shusterman et al., 2011**). The learning process can lead to a modulation of the respiration behavior in trained animals as compared to untrained animals. Moreover, it has been shown that prolonged fast respiration (>4 Hz) reduces odor-evoked $Ca^{2+}$ signals in OSNs (**Verhagen et al., 2007**). It is thus plausible that the modulation of the strength of IOS after training (**Figures 3 and 4**) could arise from changes in breathing strategy. However, we did not observe significant changes in the breathing frequency of mice when monitored at the beginning and at the end of discrimination training (**Figure 6**). The frequency was also comparable to the one previously measured in naïve awake head-restrained animals (3.1 Hz, **Patterson et al., 2013**). Likewise, IOS amplitude does not vary with the animal's breathing frequency (**Figure 7**). Therefore, the IOS potentiation we report here mostly reflects neural changes induced by the learning process rather than respiration modulations. Along this line, a similar potentiation of the odor-evoked glomeruli activity induced by fear learning has been recently reported in anesthetized animals where effects are presumably independent of respiration (**Kass et al., 2013**). Altogether these data suggest that the potential role played by breathing behavior is minor as a source of modulation of sensory input strength in olfactory associative learning tasks.

What is the behavioral relevance of this plasticity? The improved representations mediated by the training can account for discrimination accuracy and reaction time stability across a wide range of odorant dilutions. The observed increase in the sensitivity at higher dilutions of odorants tested should lower the odorant detection thresholds in mice. Interestingly, in human olfaction, trained wine tasters show lower detection thresholds and an increase in the perceived odorants intensity compare to healthy non-trained subjects (**Marino-Sanchez et al., 2010**; **Tempere et al., 2011**). Although the effect of the plasticity reported in our study lasted for 5 weeks, this enhanced representation may be lasting for longer even with brief repetitive training and may account for the long lasting perception abilities seen in professional wine tasters.

Irrespective of the molecular mechanisms, our study provides physiological evidence for the

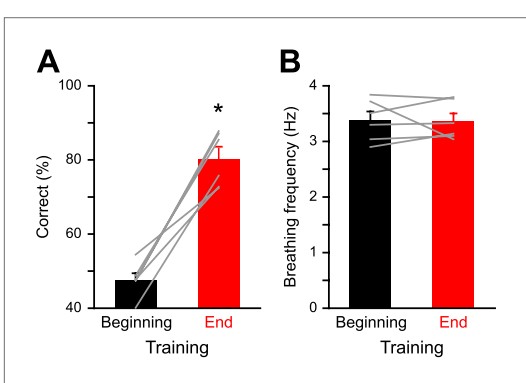

**Figure 6**. Respiration behavior is not altered by olfactory discrimination learning. (**A**) Performance levels shown by mice at the beginning (black, 300 trials averaged) and at the end of a discrimination training task (red, 300 trials averaged) with Cineol and Eugenol (*paired $t$ test, p<0.001, $n$ = 6 mice). (**B**) Average respiration frequency from the respective training blocks in (**A**). Sniff frequency remained unaltered during different learning epochs (*paired $t$ test, p=0.8, $n$ = 6 mice). Values are represented as mean ± SEM.

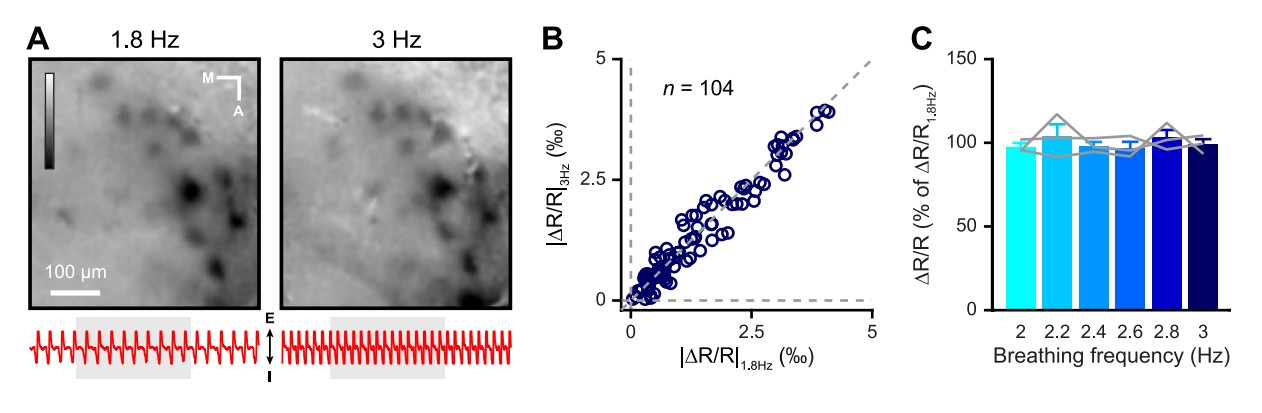

**Figure 7**. Odor-evoked intrinsic signals are independent of change in breathing frequency. (**A**) Single trial map of the amyl acetate-evoked activity reported by IOS when the mouse is breathing at 1.8 (*left*) and 3 Hz (*right*). LUT: −0.003 to 0.003. The respiration pattern recorded during each trial is shown below each image (I: inspiration, E: expiration); the light gray vertical bar represents odor presentation (5 s). (**B**) Average values of glomerular response amplitude ($\Delta R/R$) at 1.8 Hz, plotted against amplitudes at 3 Hz (Wilcoxon signed-rank test, $n$ = 104 glomeruli from three mice, p=0.2945). (**C**) Average values of glomerular response amplitude at different breathing frequencies (2, 2.2, 2.4, 2.6, 2.8 and 3 Hz). Values are normalized relative to responses recorded at 1.8 Hz (Repeated measure one-way ANOVA, $n$ = 3 mice, $F$ = 0.4499, p=0.6505). Values are represented as mean ± SEM.

existence of a functional plasticity at the sensory periphery, which helped the animal to achieve fast and accurate odor discriminations.

## Materials and methods

### Animals
All experiments were performed on adult male C57BL/6J mice (11 weeks old at the beginning of the behavioral experiments, Charles River France) in accordance with the Swiss Federal Act on Animal Protection and Swiss Animal Protection Ordinance, University of Geneva and the state of Geneva ethics committee (authorization 1007/3758/2).

### Odorants
Odorants used were iso-amyl acetate (IAA, ≥99% purity), ethyl butyrate (EB, ≥99%), 1,4-cineol (Cin, ≥85%), eugenol (Eu, ≥99%). All chemicals and mineral oil were obtained from Sigma-Aldrich (Germany) or Fluka Chemie (Germany).

### Behavioral training
All olfactory discrimination experiments were performed using four modified eight-channel olfactometers (Knosys, Lutz, FL) controlled by custom routine (kindly provided by Dr Andreas Schaefer, National Institute for Medical Research, UK) written in Igorpro (Wavemetrics, Portland, OR). Odorants were diluted from $10^0$ to $10^{-10}$ percent volume in mineral oil and further diluted 1:20 by airflow. Odorants were made freshly for each task. The task habituation training, olfactory training and reaction/discrimination time measurements were conducted as previously published (*Abraham et al., 2004*, *2010*, *2012*). In brief, a trial is initiated by breaking a light beam at the sampling port opening. This opens one of eight odor valves and a diversion valve (DV) that allows all airflow to be diverted away from the animal for 500 ms. After the release of DV, the odor is applied to the animal for 2 s. Trials were counted as correct if the animals met the criteria we set for water delivery (licking at least once in three out of four 500 ms bins) upon presentation of S+ or if licking did not occur in more than one out of four 500 ms bins for S−. For correct S+ trials mice can receive a 2–4 µl water reward at the end of 2 s stimulus period. Conversely for the incorrect S+ and correct S− trials no reward is supplied. A trial cannot be initiated unless an inter-trial interval of at least 5 s has passed. This interval was sufficiently long so that animals typically retract quickly after the end of the trial. The minimal inter-stimulus interval was

thus 5 s, which seemed to be sufficient as no habituation could be observed (performance was not correlated with the actual inter-trial interval chosen by the animal, which was around 10–20 s). No minimal sampling time was required to artificially enforce the animal to take a decision. Odors are presented in a pseudo-randomized scheme (no more than two successive presentations of the same odor). The trained group of mice was evenly distributed between the setups and the valence of odorants in a pair (S+ and S−) was switched between animals. All activated glomeruli included in our quantification had therefore an equal chance to be associated with a rewarded odor and a non-rewarded odor (*Figure 4G*).

Upon presentation of a S+ odor, the animal generally continuously breaks the beam, whereas upon presentation of an S− odor an animal familiar with the apparatus usually quickly retracts its head. Reaction times were calculated as follows: for every time point, beam breakings for S+ and S− odors were compared by bootstrapping, yielding significance value as a function of time after odor onset. The last crossing of the $p=0.05$ line determined the reaction time. In very few cases, this did not coincide with the visually identified reaction time (point of largest curvature in the log[p]-t plot) and was corrected after visual inspection.

Following the training on $10^0$ Cin vs Eu (1200 trials, *Figure 2A*), mice were trained on a pair of natural odorants (cloves vs camphor) for another experiment, which took 2 weeks. They were then trained on the different dilutions of odorants reported in this study.

For the passive exposure experiment we used the same odorant delivery system, the same number of trials and the same pseudo-randomized sequence of odorants compared to the behavior experiments. Each time, the mice were exposed in their home cage to 2 s of odor plumes at an inter-trial interval (ITI) of ~20–40 s, depending on the odorant pair and calculated from the average ITI's observed during olfactory training. During odor applications, mice were clearly investigating around the odor tube outlet. Each day 300 trials (150 rewarded and 150 non-rewarded) were presented as this was the highest number of trials finished per day by best performers.

For data presented in *Figure 6*, we performed an odor discrimination task in head-restrained mice as described previously (*Abraham et al., 2012*). Respiration was detected via a directional airflow sensor and breathing frequency was computed during the 2 s odor presentation (AWM2100V; Honeywell, Germany).

## Intrinsic optical signal imaging

For IOS imaging performed in awake head-restrained mice, the animal preparations and head post implantations were done as described previously (*Vincis et al., 2012*). For IOS imaging performed in anesthetized mice (*Figure 7*), animals were deeply anesthetized by intraperitoneal injection (i.p.) of 3.1 µl/g body weight of a mixture consisting of 60 µl Medetomidin (Dormitor, Pfizer AG, Zurich, Switzerland; 1 mg/ml), 160 Midazolam (Dormicum, Roche Pharma AG, Switzerland; 5 mg/ml) and 40 µl Fentanyl (Sintenyl, Sintetica SA, Mendrisio, Switzerland; 50 µg/ml). A local anesthetic, carbostesin (AstraZeneca, Zug, Switzerland), was subcutaneously injected before any skin incision. Anesthesia was maintained by periodic dosage (~30 µl i.p. every 30 min) of mixture containing only Midazolam (5 mg/ml) and Medetomidin (1 mg/ml). Odorants were delivered for 5 s (2 s after recording onset) using a custom made olfactometer (*Bathellier et al., 2008*; *Gschwend et al., 2012*) and images were acquired at 700 nm wavelength using the Imager 3001F system (Optical Imaging Ltd., Israel) (*Accolla et al., 2007*; *Vincis et al. 2012*). The number of repetitions for each odorant/dilution was four. The glomerulus detection procedure was done on individual time frames by drawing regions of interest (ROI). The ROI analyzed for the ΔR/R measurement, were selected based on the glomerular map obtained for each mice at the lowest dilution of four odorants used. This map was then used as the reference across different dilutions of the same odorant to calculate the ΔR/R. The same procedure was adopted for each experimental group. We excluded every region that appeared only in a single frame or that looked like blood vessels. For the experiment shown in *Figure 7*, frequency of respiration was detected via a directional airflow sensor (AWM2100V; Honeywell, Germany).

## Acknowledgements

We thank Andreas Schaefer for kindly providing the software controlling olfactometers. We thank Michael Patterson and members of the Carleton laboratory for helpful discussions and comments on the manuscript.

# Additional information

## Funding

| Funder | Grant reference number | Author |
|---|---|---|
| The Swiss National Science Foundation | PP0033_119169, PP00P3_139189 | Alan Carleton |
| The National Center of Competence in Research (NCCR) | 51AU40_125759 | Alan Carleton, Samuel Lagier |
| The European Research Council | ERC-2009-StG-243344-NEUROCHEMS | Alan Carleton |
| The Novartis Foundation for Medical Research | | Alan Carleton |
| The Carlos & Elsie de Reuter Foundation | | Alan Carleton |
| The Ernst & Lucie Schmidheiny Foundation | | Alan Carleton |
| The European Molecular Biology Organization | | Nixon M Abraham, Alan Carleton |
| The Swiss National Science Foundation | CR33I13_143723 | Ivan Rodriguez, Alan Carleton |
| The Swiss National Science Foundation | 310030E_135910 | Ivan Rodriguez |

The funders had no role in study design, data collection and interpretation, or the decision to submit the work for publication.

## Author contributions

NMA, RV, SL, Conception and design, Acquisition of data, Analysis and interpretation of data, Drafting or revising the article; IR, AC, Conception and design, Drafting or revising the article

## Ethics

Animal experimentation: All animal protocols are in accordance with the Swiss Federal Act on Animal Protection, Swiss Animal Protection Ordinance, and were approved by the University of Geneva and Geneva state ethics committees (authorization 1007/3758/2).

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
