## [Decision Letter]

Thank you for sending your work entitled “Long term functional plasticity of sensory inputs mediated by olfactory learning” for consideration at *eLife*. Your article has been favorably evaluated by a Senior editor and 3 reviewers, one of whom is a member of our Board of Reviewing Editors.

The Reviewing editor and reviewers discussed their comments before we reached this decision, and the Reviewing editor has assembled the following comments to help you prepare a revised submission.

Abraham et al. studied whether and how odor exposures to various conditions alter odor-evoked activities in the olfactory bulb as measured by optical intrinsic signal imaging (ISI). The authors found that glomerular activations were increased by paring of odors with positive and negative reinforcements compared to those evoked in naïve animals. In contrast, animals that received passive exposure did not show such a change. Furthermore, the learning-induced increase in activity lasted up to 5 weeks after training. The authors' finding that plastic changes in odor-evoked activity depend on reinforcement is very intriguing. The results are quite interesting since as the authors point out most work has focused on plasticity in central rather than peripheral processing. The pronounced increase they find in sensitivity of sensory input is therefore unanticipated. The study is thorough on several fronts. Animals undergo long-term training covering a wide range of concentrations for two different odor pairs. Maps are also tested across a concentration range, are very clear with good signal to noise, and the data are nicely quantified.

At the same time, the reviewers expressed several major related concerns, explained below:

1) Imaging in awake animals to avoid anesthesia effects is a strength, but raises questions about the source of changes in ISI responses. The authors should better address the role of altered sniffing for the trained odors. Potentiation could reflect neural processing but could also result from changes in sensory sampling by the animal. In awake animals sniff patterns vary strongly with history and expectation in both trained and untrained animals. Calcium imaging has also shown that sniffing modulates sensory input.

The discussion states that 'to our knowledge, slow signals like IOS are not sensitive to ... change of respiration'. While no evidence shows sniffing affects signals, no evidence shows that it does not. The coupling between presynaptic activity and intrinsic signals is not well understood. Wesson et al. is cited to state that fast sniffing 'is not affecting the odor evoked glomeruli activities', other papers from the Wachowiak group (Carey et al.; Verhagen et al.) do show that fast sniffing changes the temporal and spatial characteristics of maps. Neural changes are quite plausible but since they are the central conclusion of the paper it will be important to either experimentally rule out a role of sniffing, or present a more balanced discussion of the potential role of behavioral changes.

2) The passive exposure experience is not clear. It appears that the passive exposure condition involves odor plumes sent into the home cage as compared to the training condition where animals are in a testing apparatus and poking their noses into a sniff port. Is this right? I doubt that the dilutions of odors reaching the nose in these two situations are equivalent. I suspect the odors become highly diluted in the home cage, although they are likely absorbed by the bedding and so may be longer lasting. Also, mice may not sniff similarly during exposures in the passive and training conditions. Finally, the authors refer to “rewarded” and “non-rewarded” odors in the passive condition – what does this mean (perhaps they are referring to the valences in the training task and no rewards are given in the passive exposure)? Mice that were trained had a learning task, whereas the passive exposure did not involve learning of any sort - olfactory or whatever. I don't understand why they didn't have animals perform the same basic sniff and licking task, but with random rewards for both odors, or perhaps train with different odors than those used in the imaging. This would have controlled for the behavior and rewards.

3) The authors use the terms “glomeruli inputs” or “peripheral inputs” but the signal source of ISI is ambiguous compared to other techniques available more recently. The authors discuss [21] to support a presynaptic origin of ISI signals but this issue is not completely resolved. The authors should discuss about the difficulty of identifying the sources of intrinsic signals. The discussions on the mechanisms should, therefore, be more balanced with respect to signal sources.

4) The use of “discrimination time” is uncommon in literature except the authors' groups. Note that the time point at which reaction time distributions differed statistically between go vs. no-go trials does not necessarily mean that the animal discriminated with high accuracy at that time point (the accuracy can be still very low (e.g., 51%) even when the distributions of reaction times are statistically different with many trials). The authors should report the median (or mean) and the entire distribution of reaction times.

---

## [Author Response]

*1) Imaging in awake animals to avoid anesthesia effects is a strength, but raises questions about the source of changes in ISI responses. The authors should better address the role of altered sniffing for the trained odors. Potentiation could reflect neural processing but could also result from changes in sensory sampling by the animal. In awake animals sniff patterns vary strongly with history and expectation in both trained and untrained animals. Calcium imaging has also shown that sniffing modulates sensory input*.

In order to address this point, we present new data (Figure 6) in which we show no alteration in sniffing behavior before and after the learning of a similar olfactory discrimination task. Additionally, the breathing frequencies recorded in this new set of data is similar to breathing frequencies measured in non-trained “naïve” awake head-restrained animals (3.1Hz, Patterson et al., 2013).

We have now modified the Discussion to include a paragraph on breathing behavior.

*The discussion states that 'to our knowledge, slow signals like IOS are not sensitive to ... change of respiration'. While no evidence shows sniffing affects signals, no evidence shows that it does not. The coupling between presynaptic activity and intrinsic signals is not well understood. Wesson et al. is cited to state that fast sniffing 'is not affecting the odor evoked glomeruli activities', other papers from the Wachowiak group (Carey et al.; Verhagen et al.) do show that fast sniffing changes the temporal and spatial characteristics of maps. Neural changes are quite plausible but since they are the central conclusion of the paper it will be important to either experimentally rule out a role of sniffing, or present a more balanced discussion of the potential role of behavioral changes*.

We present in Figure 7 the effect of sniffing frequency on the amplitude of IOS. From 1.8Hz to 3Hz, the amplitude of IOS is constant.

Combined with the results presented in Figure 6, we conclude that the measured effect of behavioral training on IOS amplitude is unlikely to arise from changing in sniffing behavior.

We have modified the Discussion to include a paragraph on breathing behavior.

*2) The passive exposure experience is not clear. It appears that the passive exposure condition involves odor plumes sent into the home cage as compared to the training condition where animals are in a testing apparatus and poking their noses into a sniff port. Is this right? I doubt that the dilutions of odors reaching the nose in these two situations are equivalent. I suspect the odors become highly diluted in the home cage, although they are likely absorbed by the bedding and so may be longer lasting. Also, mice may not sniff similarly during exposures in the passive and training conditions. Finally, the authors refer to “rewarded” and “non-rewarded” odors in the passive condition – what does this mean (perhaps they are referring to the valences in the training task and no rewards are given in the passive exposure)? Mice that were trained had a learning task, whereas the passive exposure did not involve learning of any sort - olfactory or whatever. I don't understand why they didn't have animals perform the same basic sniff and licking task, but with random rewards for both odors, or perhaps train with different odors than those used in the imaging. This would have controlled for the behavior and rewards*.

In our freely moving behavioral setup, the odor delivery gets triggered only when the mouse is poking its head in the sampling port. In other words, without an action from the animal, no odor is presented. During a passive exposure task in the olfactometer, animals do not get a reward and thus tend to lose their interest for the task. In this context, getting a repetitive passive exposure would require either a very long time (to reach enough trials) or to select mice that are very curious (i.e., mice that do not lose interest in sampling the odor).

Likewise, we noticed that in a pseudo-randomized odor-reward association, animals tend to lose interest in odor sampling. The number of trials necessary to complete the discrimination task (4800 trials) is very large. We feared that running the same number of trials for a pseudo-randomized association with freely moving animals would not be possible with a reasonable amount of mice.

We agree with reviewer #1 that in our passive exposure paradigm, the concentration and duration of the presented odorants are most likely differing from the ones used with the trained animals. However, the paradigm we chose seemed the most feasible while remaining relevant.

An odor-reward pseudo-random association could have been used as another control experiment. A complete new set of experiments would need to be carried out to test the difference between active (non-random association) and passive (random association) exposure. Performing those tasks in head-restrained animals would help control the odor exposition in both conditions and would not depend on the animals’ motivation. Answering fully this question would be the scope of another study.

The reference to “rewarded” and “unrewarded” odor in the passive exposure context is now clarified in the main text.

*3) The authors use the terms “glomeruli inputs” or “peripheral inputs” but the signal source of ISI is ambiguous compared to other techniques available more recently. The authors discuss*
[21]
*to support a presynaptic origin of ISI signals but this issue is not completely resolved. The authors should discuss about the difficulty of identifying the sources of intrinsic signals. The discussions on the mechanisms should, therefore, be more balanced with respect to signal sources*.

We do have experimental evidence (manuscript submitted) showing that odor-evoked IOS originate from presynaptic inputs (OSN) in the olfactory bulb. Extending on Gurden article, we show not only that IOS do not depend on neuronal post-synaptic activity, but we also show that astrocytes do not contribute to IOS. We thus confirm the exclusive pre-synaptic origin of odor evoked intrinsic signals in the OB.

*4) The use of “discrimination time” is uncommon in literature except the authors' groups. Note that the time point at which reaction time distributions differed statistically between go vs. no-go trials does not necessarily mean that the animal discriminated with high accuracy at that time point (the accuracy can be still very low (e.g., 51%) even when the distributions of reaction times are statistically different with many trials). The authors should report the median (or mean) and the entire distribution of reaction times*.

In the Result and Discussion sections of the main text we referred to “reaction time” while in the Methods we used a couple of times the term “discrimination time”. In the Methods, we now use the term “reaction/discrimination time” while referring to the previously published studies. When the animals perform with high accuracies, the discrimination time involve the comparison of reaction time measurements for both rewarded and non-rewarded odor. In the Results section as we report the time measurements when the animals perform with different accuracies we now use the term “reaction times” for the sake of clarity. The median of the distributions of reaction times is shown in Figure 2.